# Enhanced Authenticated Key Agreement for Surgical Applications in a Tactile Internet Environment

**DOI:** 10.3390/s22207941

**Published:** 2022-10-18

**Authors:** Tian-Fu Lee, Xiucai Ye, Wei-Yu Chen, Chi-Chang Chang

**Affiliations:** 1Department of Medical Informatics, Tzu Chi University, No. 701, Zhongyang Road, Sec. 3, Hualien 970, Taiwan; 2Department of Computer Science, University of Tsukuba, Tsukuba 3058577, Japan; 3Department of Medical Informatics, Chung Shan Medical University, No. 110, Section 1, Jianguo North Road, South District, Taichung City 402, Taiwan; 4Department of Information Management, Ming Chuan University, No. 5 De Ming Rd., Taoyuan City 333, Taiwan

**Keywords:** Tactile Internet, 5G, authentication, key agreement, surgery, robotic arm

## Abstract

The Tactile Internet enables physical touch to be transmitted over the Internet. In the context of electronic medicine, an authenticated key agreement for the Tactile Internet allows surgeons to perform operations via robotic systems and receive tactile feedback from remote patients. The fifth generation of networks has completely changed the network space and has increased the efficiency of the Tactile Internet with its ultra-low latency, high data rates, and reliable connectivity. However, inappropriate and insecure authentication key agreements for the Tactile Internet may cause misjudgment and improper operation by medical staff, endangering the life of patients. In 2021, Kamil et al. developed a novel and lightweight authenticated key agreement scheme that is suitable for remote surgery applications in the Tactile Internet environment. However, their scheme directly encrypts communication messages with constant secret keys and directly stores secret keys in the verifier table, making the scheme vulnerable to possible attacks. Therefore, in this investigation, we discuss the limitations of the scheme proposed by Kamil scheme and present an enhanced scheme. The enhanced scheme is developed using a one-time key to protect communication messages, whereas the verifier table is protected with a secret gateway key to mitigate the mentioned limitations. The enhanced scheme is proven secure against possible attacks, providing more security functionalities than similar schemes and retaining a lightweight computational cost.

## 1. Introduction

The fifth generation (5G) network provides fast speeds, high data rates, very low latency, and reliable connections for intelligent devices, sensors, and actuators, as well as the ability to communicate through a single device, such as a smartphone. When 5G technology matures, it will provide 100 Gbps coverage, 10 GB/s peak data rates, and more than 100 billion smart device connections to the entire Internet of Things [1]. The high capacity and speed of the 5G network will provide many opportunities for the IoT environment. The Tactile Internet (TI) represents a future development goal with respect to the Internet of Things (IoT), including human–machine interaction and machine–machine interaction, which will enable real-time collaboration and innovative applications in the industrial, social, and commercial fields of the Internet [2,3].

The Tactile Internet will use 5G URLLC (ultra-reliable and low-latency communication) functionality to provide users with ultra-fast Internet so that haptic interaction can be realized through visual feedback [3]. This visual feedback relates to audio–visual interaction, real-time control of robotic systems and actuators, and real-time control of the human body and the environment around it. With the increasing availability of high-speed Internet connections, such low-latency functions will lead to enhanced human–machine (tactile) interactions that can be transmitted to the other end of the world in real time [1,3,4]. However, such messages may face security or performance risks once they are transmitted. Therefore, any unauthorized access may lead to an unplanned or unexpected surgery, which could lead to adverse consequences or even death.

The open nature of Tactile Internet connections makes them vulnerable to a variety of security attacks, including replay, denial of service, man-in-the-middle, differential privacy, error data injection, impersonation, and modification attacks, as well as malicious software attacks, requiring secure Tactile Internet access. The remote surgery application establishes a secure user authentication protocol, which allows authorized and registered surgeons to authenticate each other and to generate a shared secure session key for secure and reliable communications with others.

### 1.1. The Model of a Tactile Internet Remote Surgery Application

Figure 1 illustrates a simple model of a Tactile Internet remote surgery application. A hospital operating room includes robotic arms with tactile sensors and actuators; gateways, such as access points (APs); and patients to be operated on. A remote surgeon controls the robotic arm using instructions provided by a mobile device (or multiple mobile devices) and receives the results of the operation on the screen. All devices must be registered with a trusted institution (TA).

### 1.2. Related Works

The Tactile Internet can allow doctors to perform accurate, remote surgery more urgently than ever before. The transmission of the data would require the surgical manipulator to move the scalpel with a delay of less than 1 ms to allow the scalpel to move in the correct direction. To obtain the real-time status of the patient, high-resolution organ images and medical equipment data must also be sent back to doctors within 1 ms. Recently, many authenticated key agreement approaches have been developed for remote medical systems. For example, in 2018, Amin et al. [5] proposed a robust and anonymous patient monitoring system based on wireless medical sensor networks to provide secure access to patient data in WMSN environments. In the same year, Wu et al. [6] developed a lightweight and robust authentication scheme for personalized healthcare systems using wireless medical sensor networks and demonstrated that their scheme meets common security requirements and prevents attackers from tracking users. Using wireless medical sensor networks, Chandrakar [7] presented a secure remote user authentication protocol for healthcare monitoring that provides privacy, data security, and user authentication to access real-time health information over an insecure channel. Kaur et al. [8] presented a protocol in 2020 that provides the surgeon, robotic arm, and trusted authority (TA) with secure communications, leveraging the advantages of elliptic curve cryptography (ECC) and biometrics. In 2020, Nykvist et al. [9] developed and implemented a lightweight, portable IDS over wireless networks and evaluated throughput, power consumption, and response time. In 2021, Bolton et al. [10] discussed and considered potential data security and privacy issues that may arise when large amounts of data are processed and stored in the cloud. Additional research on the use of the Tactile Internet in remote surgery [8,11,12] provides important background information about the use of the Tactile Internet in remote surgery. For example, Wazid et al. [12] presented a generalized authentication model that can be used to perform authentication among communicating parties to ensure secure remote surgery in the TI environment. In 2021, Kamil et al. [11] proposed an authentication and key agreement (AKA) scheme for a Tactile Internet remote surgery application using lightweight cryptographic operations, such as the one-way hash function and bitwise exclusive OR (XOR), making the scheme ultra-lightweight and suitable for the Tactile Internet environment. However, the proposed scheme directly encrypts communication messages with the constant secret keys of the remote surgeon and the long-life secret key of the robotic arm, directly storing secret keys of the robotic arm in the gateway database; therefore, the scheme cannot resist robotic arm compromise attacks and stolen verifier attacks. Additionally, the scheme proposed by Kamil et al. misuses exclusive OR operations, preventing its correct execution.

### 1.3. Our Motivation

Many AKA schemes have been recently developed for a Tactile Internet for remote surgery. However, most of these schemes are subject to limitations in terms of security and efficiency. Performance improvement and security considerations are two major factors associated with the Tactile Internet because inappropriate and insecure authentication key agreements for the Tactile Internet may cause misjudgment and improper operation by medical staff, endangering the life of patients.

### 1.4. Our Contributions

In this investigation, we discuss the limitations of the scheme proposed by Kamil et al., including the failure to resist potential attacks and incorrect execution. In order to overcome these limitations, we investigation develop an enhanced authenticated key agreement scheme based on the scheme proposed by Kamil et al. for the Tactile Internet environment. The enhanced scheme adopts a one-time key to protect communication messages such that the adversary cannot derive valuable information from previous messages and protects secret keys of robotic arms with a secret gateway key. Thus, the enhanced scheme requires more computations and response time than the protocol proposed by Kamil et al. However, the enhanced scheme solves the previous limitations, provides improved functionality, and retains a low computational cost. The contributions of this study are summarized as follows.

1. In this investigation, we develop an efficient and secure authenticated key agreement scheme based on the scheme proposed by Kamil et al. for the Tactile Internet environment.

2. The enhanced scheme adopts a one-time key to protect communication messages and stores the secret keys of robotic arms, which are encrypted the secret gateway key, in the gateway database to overcome the limitations of the previous scheme.

3. Burrows–Abadi–Needham (BAN) logic provides mutual authentication and session key security through its authentication proof. The heuristic security analyses of the enhanced scheme are presented to verify other security requirements.

4. Compared with related schemes, the enhanced scheme avoids the limitations of pervious schemes, providing improved security properties and retaining low computational cost.

### 1.5. Organization of Paper

The rest of the paper is organized as follows. In Section 2, we introduce the scheme proposed by Kamil et al. and discuss its weaknesses. In Section 3, we introduce an enhanced authenticated key agreement scheme for the Tactile Internet environment. In Section 4, we analyze the security and performance of the enhanced scheme. Finally, in Section 5, we present our conclusions.

## 2. Preliminary

In this section, we review the authentication and key agreement scheme proposed by Kamil et al. and discuss its limitations. The notations used in this paper are elaborated in Table 1.

### 2.1. Review of the Scheme of Kamil et al.

In 2020, Kamil et al. [11] proposed an authentication and key agreement scheme using the Tactile Internet for remote surgery. Prior to the announcement, they discussed Tactile Internet technology in remote surgery, the potential of network architecture for the Internet of Thing (IoT), and the security issues of Tactile Internet technology in remote surgery.

The scheme proposed by Kamil et al. comprises four entities: a trusted authority (TA), remote surgeons, gateways, and robotic arms. Gateways act as system administrators and serve as central authentication points. Without BS, other entities would never be able to trust each other in the authentication and key agreement scheme. Kamil et al.’s scheme consists of the following phases: registration of the gateway and robotic arm, registration of the user, the authentication and key agreement phase, the password update phase, the addition of the dynamic robotic arm, and the revocation phase.

**Table 1 sensors-22-07941-t001:** Notations.

Notation	Description
TA	Trusted authority
$G_{i}$	Gateway i
$RM_{j}$	Robotic arm
$S_{k}$	Remote surgeon
$RID_{i}$ $RID_{j}$	Identity of gateway Identity of robotic arm
$RID_{k}$	$\mathrm{Identity}\;\mathrm{of}\;S_{k}$
‖	Concatenation operation
$TS_{x}$	Timestamp at instant
ΔT	Allowable network transmission delay x
⊕	Bitwise exclusive OR (XOR) operation
h(.)	Hash function
K	Session key
PW	$\mathrm{Password}\;\mathrm{of}\;S_{k}$
A	Adversary

#### 2.1.1. Gateway and Robotic Arm Registration Phase

Before placing the gateway and robot (or robotic arm) in the hospital operating room, they must register with the TA. These devices are generated and preloaded with secrets. The registration process is performed by the TA through the following steps.

Step 1: $\;TA\Rightarrow G_{i}:\;M_{1}=\left(RID_{i},D_{i},RID_{j},D_{j}\right)$.

The trust authority (TA) first chooses a unique identity ($RID_{TA}$) and a one-way hash function operation ($h:{\left\{0,1\right\}}^{*}\to Z_{q}^{*}$) for itself. Next, the TA chooses $RID_{i}$ and $RID_{j}$ as the identities of the gateway ($G_{i}$) and a robotic arm ($RM_{j}$), respectively, picks a secret ($s\in Z_{q}^{*}$), and computes $D_{i}=h\left(s,RID_{TA},RID_{i}\right)$ and $D_{j}=h\left(s,RID_{TA},RID_{j}\right)$. Finally, the TA stores $\left(RID_{i},D_{i},RID_{j},D_{j}\right)$ and sends $M_{1}$ to $G_{i}$ through a secure channel.

Step 2: Gi⇒RMj: M2=(RIDj,Dj).

After gateway $G_{i}$ receives $M_{2}$, $G_{i}$ stores $\left(RID_{i},D_{i},RID_{j},D_{j}\right)$ and sends $M_{2}$ to $RM_{j}$.

#### 2.1.2. User Registration Phase

In this stage, when the remote surgeon wants to use the robotic arm for remote surgery, they first need to register with the TA. The process is as follows.

Step 1: $S_{k}\Rightarrow TA:\;M_{3}=\left(D_{k},HPW_{k}\right)$.

The remote surgeon ($S_{k}$) first picks an identity ($RID_{k}$), a password ($PW_{k}$), and a random nonce ($B_{k}$) and computes $D_{k}=h\left(RID_{k},B_{k}\right)$ and $HPW_{k}=h\left(PW_{k},B_{k}\right)$. Next, $S_{k}$ sends $M_{3}$ to the TA using a secure channel.

Step 2: $TA\Rightarrow S_{k}:\;M_{4}=\left(\alpha ,\beta ,h\left(.\right)\right)$.

When the TA receives $M_{3}$, the TA at first picks a random C and then computes $\alpha =h\left(C,D_{i}\right)⊕h\left(D_{k},HPW_{k}\right)$ and $\beta =C⊕h\left(RID_{i},D_{i}\right)$. After the TA stores (α,β,h(.)) into the memory of a mobile device, the TA sends the mobile device to the surgeon through a secure channel.

Step 3: Store $\left(A_{1},A_{2},h\left(.\right)\right)$ in smart card.

When $S_{k}$ receives the mobile device, $S_{k}$ uses a smart card to compute $A_{1}=h\left(PW_{k},RID_{k}\right)⊕B_{k}$ and $A_{2}=h\left(B_{k},HPW_{k},D_{k}\right)$. Next, $S_{k}$ stores $A_{1}$ and $A_{2}$ in the smart card.

#### 2.1.3. User Login Phase

First, $S_{k}$ must input his/her identity or password into the mobile device in order to access the service of robotic arms for remote surgery. Upon successful verification, the mobile device sends a login request message to the gateway ($G_{i}$). The login process is as follows.

$S_{k}$ first inputs his identity ($RID_{k}$) and password ($PW_{k}$) and computes $B_{k}=A_{1}⊕h\left(PW_{k},RID_{k}\right)$, $D_{k}=h\left(RID_{k},B_{k}\right)$, $HPW_{k}=h\left(PW_{k},B_{k}\right)$, and $A_{2}^{*}=h\left(B_{k},HPW_{k},D_{k}\right)$ to verify $A_{2}$. The mobile device checks whether $A_{2}^{*}$ is the same as the $A_{2}$. If so, the identity and password of the surgeon are verified by the smart card. Otherwise, the session is aborted.

#### 2.1.4. Authentication and Key Agreement Phase

In this phase, in order to perform remote surgery in an emergency, the remote surgeon needs to use the robotic arm to perform remote surgery on the patient through the authorization of the gateway device. The mutual authentication and key agreement process of the scheme proposed by Kamil et al. is described as follows.

Step 1: $\;S_{k}\to G_{i}:\;M_{1}=(A_{4},A_{5},A_{6},TS_{1})$.

The mobile device of the remote surgeon ($S_{k}$) first picks a random nonce ($R_{k}$) and a timestamp ($TS_{1}$) and computes $A_{3}=\alpha ⊕h\left(D_{k},HPW_{k}\right)$, $A_{4}=\beta ⊕TS_{1}$, $A_{5}=h\left(R_{k},A_{3},TS_{1}\right)$, and $A_{6}=(R_{k}\|A_{5})⊕A_{3}$. Next, the remote surgeon sends a login request message ($M_{1}$) to $G_{i}$.

Step 2: $\;G_{i}\to RM_{j}:M_{2}=\left(A_{7},A_{8},A_{9}\right)$.

After $G_{i}$ receives the authentication request message ($M_{1}$), $G_{i}$ computes $C^{*}=A_{4}⊕h\left(RID_{i},D_{i}\right)⊕TS_{1}$ using the identity of gateway $RID_{i}$ and $D_{i}$ ($A_{3}^{*}=h\left(C^{*},D_{i}\right)$) and computes $R_{k}^{*}\;\|\;A_{5}=A_{6}⊕A_{3}^{*}$ to obtain the random number ($R_{k}^{*}$) of the remote surgeon. Then, $G_{i}$ checks the freshness of the message by verifying whether $TR_{1}-TS_{1}\leq \Delta T$, where $TR_{1}$ is the time at which the message is received, $TS_{1}$ is the time at which it was sent, and ΔT is the transmission delay. If the timestamp is legal, $G_{i}$ computes $A_{5}^{*}=h\left(R_{k}^{*},A_{3}^{*},TS_{1}\right)$ to verify whether the $A_{5}^{*}$ is the same as $A_{5}$. If the verification is successful, the surgeon ($S_{k}$) is authenticated by $G_{i}$. Then, $G_{i}$ chooses a random nonce ($R_{i}$) and a timestamp ($TS_{2}$) and computes $A_{7}=C^{*}⊕h\left(RID_{j},D_{j},R_{i},R_{k}^{*},TS_{2}\right)$,$A_{8}=D_{j}⊕\left(R_{i}\;\left|\left|\;R_{k}^{*}\;\right|\right|TS_{2}\right)$, and $A_{9}=h\left(RID_{j},D_{j},C^{*},R_{i},TS_{2}\right)$. Finally, $G_{i}$ sends $M_{2}$ to the robotic arm ($RM_{j}$).

Step 3: $RM_{j}\to G_{i}:\;M_{3}=\left(A_{10},A_{11}\right)$.

Upon receiving the tuple $\left(A_{7},A_{8},A_{9}\right)$, $RM_{j}$ computes $R_{i}^{*}\;\left|\left|\;R_{k}^{**}\;\right|\right|TS_{2}=A_{8}⊕D_{j}$ to obtain the random numbers $R_{i}^{*}$ and $R_{k}^{**}$, where $R_{i}^{*}$ belongs to the gateway and $R_{k}^{**}$ belongs to the remote surgeon, and checks the freshness of the message by verifying whether $TR_{2}-TS_{2}\leq \Delta T$, where $TR_{2}$, $TS_{2}$, and ΔT are the time at which the message was sent, the arrival time of the message, and the transmission delay, respectively. If the freshness of timestamp is verified, $RM_{j}$ computes $C^{**}=A_{7}⊕h\left(RID_{j},D_{j},R_{i}^{*},R_{k}^{**},TS_{2}\right)$ and $A_{9}^{*}=h\left(RID_{j},D_{j},C^{**},R_{i}^{*},TS_{2}\right)$. Finally, $RM_{j}$ verifies whether $A_{9}^{*}$ is the same as $A_{9}$. If verification is successful, the gateway is authenticated by $RM_{j}$. Next, $RM_{j}$ chooses a random number ($R_{j}$) and a timestamp ($TS_{3}$) and computes the session key $K_{1}=h\left(R_{i}^{*},R_{k}^{**},R_{j}\right)$, $A_{10}=h\left(R_{i}^{*},R_{j},K_{1},RID_{j},D_{j},TS_{3}\right)$, and $A_{11}=R_{i}^{*}⊕(R_{j}\;\|\;TS_{3})$. Finally, $RM_{j}$ sends $M_{3}$ to $G_{i}$ through a public channel.

Step 4: $\;G_{i}\to S_{k}:\;M_{4}=\left(A_{8},A_{12},A_{13}\right)$.

When $G_{i}$ receives $M_{3}$, $G_{i}$ computes $R_{j}^{*}\;\|\;TS_{3}=A_{11}⊕R_{i}$ to obtain the random number of $RM_{j}$, using the random number of $G_{i}$ and timestamp $TS_{3}$, and checks the freshness of the message by verifying whether $TR_{3}-TS_{3}\leq \Delta T$, where $TR_{3}$, $TS_{3}$, and ΔT are the time at which the message was sent, the arrival time of the message, and the transmission delay, respectively. If the freshness of the timestamp is legal, $G_{i}$ computes the session key $K_{2}=h\left(R_{i},R_{k}^{*},R_{j}^{*}\right)$ and $A_{10}^{*}=h\left(R_{i},R_{j}^{*},K_{2},RID_{j},D_{j},TS_{3}\right)$. $G_{i}$ checks whether $A_{10}^{*}$ is the same as $A_{10}$. If so, the robotic arm ($RM_{j}$) is authenticated by $G_{i}$. Next, $G_{i}$ computes $A_{12}=h\left(K_{2},R_{i},R_{j}^{*},A_{8},TS_{4}\right)$ and $A_{13}=(R_{i}\left|\left|R_{j}^{*}\right|\right|TS_{4})⊕R_{k}^{*}$ and sends $M_{4}$ to $S_{k}$, where $TS_{4}$ is the timestamp.

Step 5: Verification of the remote surgeon.

When $S_{k}$ receives $M_{4}$, $S_{k}$ first computes $R_{i}^{*}\;\left|\left|\;R_{j}^{**}\;\right|\right|\;TS_{4}=A_{13}⊕R_{k}$ using the random number ($R_{k}$) and then checks the freshness of the message by verifying whether $TR_{4}-TS_{4}\leq \Delta T$, where $TR_{4}$, $TS_{4}$, and ΔT are the time at which the message was sent, the arrival time of the message, and the transmission delay, respectively. If the timestamp is fresh, $S_{k}$ computes the session key $K_{3}=h\left(R_{i}^{*},R_{j}^{**},R_{k}\right)$ and $A_{12}^{*}=h\left(K_{3},R_{i}^{*},R_{j}^{**},A_{8},TS_{4}\right)$ to verify $A_{12}$. If the verification is successful, $G_{i}$ and $RM_{j}$ are authenticated by $S_{k}$.

The mutual authentication of the remote surgeon and the robotic arm requires the assistance of the gateway for remote authentication. Additionally, secure communication during remote surgery is achieved with the secret session key, $K=K_{1}=K_{2}=K_{3}$.

#### 2.1.5. Password Updating Phase

In this phase, when the remote surgeon thinks that his password has been leaked, for security reasons, he can change his password at any time. The password renewal phase is as follows.

The remote surgeon ($S_{k}$) inputs his original password ($PW_{k}^{*}$) and identity ($RID_{k}^{*}$) into the mobile device, and the mobile device computes $B_{k}^{*}=A_{1}⊕h\left(PW_{k}^{*},RID_{k}^{*}\right)$, $HPW_{k}^{*}=h\left(PW_{k}^{*},B_{k}^{*}\right)$, $D_{k}^{*}=h\left(RID_{k}^{*},B_{k}^{*}\right)$, and $A_{2}^{**}=h\left(B_{k}^{*},HPW_{k}^{*},D_{k}^{*}\right)$ to check whether $A_{2}^{**}$ is the same as $A_{2}$. If the verification is successful, the password and identity of the surgeon are verified. Next, the card reader prompts $S_{k}$ to input a new password ($PW_{k}^{new}$) and a nonce ($B_{k}^{new}$). Then, it computes $HPW_{k}^{new}=h\left(PW_{k}^{new},B_{k}^{new}\right)$, $D_{k}^{new}=h\left(RID_{k},B_{k}^{new}\right)$, $A_{1}^{new}=h\left(PW_{k}^{new},RID_{k}\right)⊕B_{k}^{new}$, $A_{2}^{new}=h\left(B_{k}^{new},HPW_{k}^{new},D_{k}^{new}\right)$, and $\alpha^{new}=\alpha ⊕h\left(D_{k}^{*},HPW_{k}^{*}\right)⊕h\left(D_{k}^{new},HPW_{k}^{new}\right)$. Finally, the mobile device replaces α, $A_{1}$, and $A_{2}$, with $\alpha^{new}$, $A_{1}^{new}$, and $A_{2}^{new}$, respectively.

#### 2.1.6. Dynamic Robotic Arm Addition Phase

After placing these robotic arms in the operation room, additional robots may be required for improved service delivery. The following steps are required.

The TA first chooses a new identity ($RID_{j}^{+}$) and computes $D_{j}^{+}=h\left(s,RID_{TA},RID_{j}^{+}\right)$. The TA stores $\left(RID_{j}^{+},\;D_{j}^{+}\right)$ in the memory of the new robotic arm and sends the tuple to the gateway ($G_{i}$) through a secure channel. When $G_{i}$ receives the tuple $\left(RID_{j}^{+},\;D_{j}^{+}\right)$, $G_{i}$ stores it in its repository.

#### 2.1.7. Revocation Phase

When the remote surgeon’s mobile device is stolen by an attacker, the attacker can reuse the data from the mobile device, thus impersonating the legitimate doctor. The same method is applied to the robot arm; the attacker can analyze the sensitive information in the robotic arm and compute the session key to execute an attack. In addition, attackers can swap out a robotic arm with a cloned robotic arm, which can lead to life-threatening conditions in patients who require medical attention. The proposed scheme involves two revocation processes: revocation of compromised mobile devices and revocation of compromised robotic arms.

1. Revocation of Smart Card: Steps can be taken to prevent compromised mobile devices from gaining access to the network. The TA first chooses a new identity ($RID_{i}^{new}$) and computes $D_{i}^{new}=h\left(s,RID_{TA},RID_{i}^{new}\right)$. Next, the TA sends the tuple $\left(RID_{i}^{new},D_{i}^{new}\right)$ to $G_{i}$. When $G_{i}$ receives $\left(RID_{i}^{new},D_{i}^{new}\right)$, $G_{i}$ replaces $\left(RID_{i},D_{i}\right)$ with $\left(RID_{i}^{new},D_{i}^{new}\right)$ and stores it in its database.

2. Revocation of Robotic Arm: Suppose $RID_{j}$ is the identity of the malicious or compromised robot. In order to prevent the malicious or damaged robotic arm from being verified by the remote surgeon and accessing the network, the following steps are performed in order to log off the manipulator. The TA computes $\Pi =\left(RID_{j}\;||\;D_{j})⊕h(RID_{i},D_{i}\right)$ and sends $\left(\Pi ,r{\mathrm{ev}}_{req}\right)$ to $G_{i}$, where $r{\mathrm{ev}}_{req}$ is the revocation request. When $G_{i}$ receives the tuple $\left(\Pi ,r{\mathrm{ev}}_{req}\right)$, $G_{i}$ computes $RID_{j}\;\|\;D_{j}=\Pi ⊕h(RID_{i},D_{i})$. Finally, $G_{i}$ deletes the tuple $\left(RID_{i},D_{i}\right)$ from its database.

### 2.2. Limitations of the Authenticated Key Agreement Proposed by Kamil et al.

The authenticated key agreement scheme proposed by Kamil et al. directly encrypts communication messages between the gateway and the remote surgeon with the constant secret keys of the remote surgeon and directly encrypts communication messages between the gateway and the robot arm with the long-life secret key of the robotic arm so that an attacker who has captured a robotic arm can derive secret keys of the remote surgeon from previous messages and successfully impersonate the remote surgeon and the robotic arm. The attacker can successfully compute session keys from previous messages to decrypt communication messages between the remote surgeon, the gateway, and the robotic arm to trick legal participants. Additionally, the scheme of Kamil et al. directly stores secret keys of robot arms, so an attacker who has stolen the verifier table can successfully impersonate the robot arm. Accordingly, the scheme proposed by Kamil et al. cannot resist robotic arm compromise attacks and stolen verifier attacks. Moreover, the scheme proposed by Kamil et al. misuses exclusive OR operations, preventing its correct execution.

Below, we discuss the limitations of the scheme proposed by Kamil et al. in detail.

#### 2.2.1. Failure to Resist Robotic Arm Compromise Attacks

1. Scenario I: Impersonation of a surgeon.

In the scheme proposed by Kamil et al., when a robotic arm ($RM_{j}$) is compromised, an attacker (A) can obtain $RID_{j}$ and $D_{j}$. The attacker (A) obtains $A_{8}$ from previous communication messages and computes $R_{i}\left|\left|R_{k}\right|\right|TS_{2}=A_{8}⊕D_{j}$ to obtain the random secrets ($R_{i}$) of the gateway ($G_{i}$) and $R_{k}$ of the remote surgeon ($S_{k}$). Next, A computes $C=A_{7}⊕h(RID_{j},D_{j},R_{i},R_{k},TS_{2})$ to obtain the random secret (C) of TA. A obtains previous communication messages ($A_{4}$,$A_{5}$,$\;A_{6}$,$TS_{1})$ of $S_{k}$ and computes $\beta =A_{4}⊕TS_{1}$, $A_{3}=(R_{k}||A_{5})⊕A_{6}\left(=h(C,D_{i}\right))$. A can compute $\tilde{A_{4}}=\beta ⊕TS_{1}^{*}$, $\tilde{A_{5}}=h\left(\tilde{R_{k}},A_{3},\tilde{TS_{1}}\right)$ and $\tilde{A_{6}}=\tilde{R_{k}}||\tilde{A_{5}}⊕A_{3}$ and send out a service request ($\tilde{M_{1}}=\left(\tilde{A_{4}},\tilde{A_{5}},\tilde{A_{6}},\tilde{TS_{1}}\right)$) to impersonate $S_{k}$, where $\tilde{R_{k}}$ is a nonce selected by A, and $\tilde{TS_{1}}$ is the current timestamp.

Upon receiving $M_{4}=\left(A_{8},A_{12},A_{13}\right)$ form $G_{i}$, A can compute $R_{i}^{*}\;\left|\left|\;R_{j}^{**}\;\right|\right|\;TS_{4}=A_{13}⊕\tilde{R_{k}}$ and the session key ($K_{3}=h\left(R_{i}^{*},R_{j}^{**},\tilde{R_{k}}\right)$) shared with $G_{i}$ and $RM_{j}$ and successfully impersonate the surgeon ($S_{k}$). Therefore, the scheme proposed by Kamil et al. fails to resist robotic arm compromise attacks.

2. Scenario II: Impersonation of a gateway.

According to the analyses of Scenario I, the attacker (A) can easily derive $A_{3}\left(=h(C,D_{i}\right))$, the random secret (C) from previous communication messages. Upon receiving $M_{1}=\left(A_{4},A_{5},\;A_{6},TS_{1}\right)$ from $S_{k}$, A computes $h\left(RID_{i},D_{i}\right)=A_{4}⊕C⊕TS_{1}$ and $R_{k}^{*}\;||\;A_{5}=A_{6}⊕A_{3}$. Then, A chooses a nonce ($\tilde{R_{i}}$) and picks the current timestamp ($\tilde{TS_{2}})$ and then computes $\tilde{A_{7}}=C⊕h\left(RID_{j},D_{j},\tilde{R_{i}},R_{k}^{*},\tilde{TS_{2}}\right)$, $\tilde{A_{8}}=D_{j}⊕\left(\tilde{R_{i}}\;\left|\left|\;R_{k}^{*}\;\right|\right|\tilde{TS_{2}}\right)$, and $\tilde{A_{9}}=h\left(RID_{j},D_{j},C,\tilde{R_{i}},\tilde{TS_{2}}\right)$ and sends $\tilde{M_{2}}=\left(\tilde{A_{7}},\tilde{A_{8}},\;\tilde{A_{9}}\right)$ to $RM_{j}$.

Upon receiving $M_{3}=\left(A_{10},A_{11}\right)$, A computes $R_{j}^{*}\;||\;TS_{3}=A_{11}⊕\tilde{R_{i}}$ and the session key ($K_{2}=h\left(\tilde{R_{i}},R_{k}^{*},R_{j}^{*}\right)$) shared with $G_{i}$ and $RM_{j}$. Next, A computes $\tilde{A_{12}}=h\left(K_{2},\tilde{R_{i}},R_{j}^{*},\tilde{A_{8}},\tilde{TS_{4}}\right)$ and $\tilde{A_{13}}=\left(\tilde{R_{i}}\left|\left|R_{j}^{*}\right|\right|\tilde{TS_{4}}\right)⊕R_{k}^{*}$, and sends $\tilde{M_{4}}=\left(\tilde{A_{8}},\tilde{A_{12}},\;\tilde{A_{13}}\right)$ to $S_{k}$, where $\tilde{TS_{4}}$ is the current timestamp. A successfully impersonates the gateway ($G_{i}$); therefore, the scheme proposed by Kamil et al. fails to resist robotic arm compromise attacks.

3. Scenario III: Violation of session key security.

According to the analyses of Scenario I, the attacker (A) can easily derive $A_{3}\left(=h(C,D_{i}\right))$, the random secret (C) from previous communication messages. First, A impersonate $S_{k}$ to compute $\tilde{A_{4}}=\beta ⊕TS_{1}^{*}$, $\tilde{A_{5}}=h\left(\tilde{R_{k}},A_{3},\tilde{TS_{1}}\right)$, and $\tilde{A_{6}}=\tilde{R_{k}}||\tilde{A_{5}}⊕A_{3}$, and to send a service request ($\tilde{M_{1}}=\left(\tilde{A_{4}},\tilde{A_{5}},\tilde{A_{6}},\tilde{TS_{1}}\right)$) to $G_{i}$, where $\tilde{R_{k}}$ is a nonce selected by A, and $\tilde{TS_{1}}$ is the current timestamp.

Then, A eavesdrops on communications between $G_{i}$ and another robotic arm ($RM_{j}^{\prime }$) and obtains $M_{2}=\left(A_{7},A_{8},A_{9}\right)$ and $M_{3}=\left(A_{10},A_{11}\right)$, where $RID_{j}^{\prime }$ is the identity of $RM_{j}^{\prime }$, $\;D_{j}^{\prime }$ is the secret key of $RM_{j}^{\prime }$, $A_{7}=C^{*}⊕h\left(RID_{j}^{\prime },D_{j}^{\prime },R_{i},\tilde{R_{k}},TS_{2}\right)$, $A_{8}=D_{j}^{\prime }⊕\left(R_{i}\;\left|\left|\;\tilde{R_{k}}\;\right|\right|TS_{2}\right)$, $A_{9}=h\left(RID_{j}^{\prime },D_{j}^{\prime },C^{*},R_{i},TS_{2}\right)$, $A_{10}=h\left(R_{i}^{*},R_{j},K_{1},RID_{j},D_{j},TS_{3}\right)$, and $A_{11}=R_{i}^{*}⊕(R_{j}\;||\;TS_{3})$. Upon receiving $M_{4}=\left(A_{8},A_{12},A_{13}\right)$ from $G_{i}$, where $A_{12}=h\left(K_{2},R_{i},R_{j}^{*},A_{8},TS_{4}\right)$ and $A_{13}=(R_{i}\left|\left|R_{j}^{*}\right|\right|TS_{4})⊕\tilde{R_{k}}$, A can compute $R_{i}^{*}\;\left|\left|\;R_{j}^{**}\;\right|\right|\;TS_{4}=A_{13}⊕\tilde{R_{k}}$ and the secret key of $RM_{j}^{\prime }$, $D_{j}^{\prime }=A_{8}⊕\left(R_{i}^{*}\left|\left|\;\tilde{R_{k}}\;\right|\right|TS_{2}\right)$.

Although the attacker (A) does not have $RM_{j}^{\prime }$’s identity ($RID_{j}^{\prime }$), A can still monitor other communications between $S_{k}$, $G_{i}$, and some robotic arms ($RM_{j}^{*}$). A computes $(R_{1}\;\|\;R_{2}\;||TS_{2})=\left(A_{8}⊕D_{j}^{\prime }\right)$ and verifies whether $TS_{2}$ is a current timestamp. If successful, A makes sure that $RM_{j}^{*}$ is $RM_{j}^{\prime }$ and $R_{1}$ is $R_{i}$ from $G_{i}$ and that $R_{2}$ is $R_{k}$ from $S_{k}$. Then, A computes $(R_{i}\left|\left|R_{j}\right|\right|TS_{4})=A_{13}⊕R_{k}$. Accordingly, A can obtain the session key ($K=h\left(R_{i},R_{k},R_{j}\right)$) of $S_{k}$, $G_{i}$, and $RM_{j}^{\prime }$ to decrypt communication messages between $S_{k}$, $G_{i}$, and $RM_{j}^{\prime }$ to perform man-in-the-middle attacks and modification attacks and to trace $RM_{j}^{\prime }$.

#### 2.2.2. Failure to Resist Stolen Verifier Attacks

In the register phase of the scheme proposed by Kamil et al., the gateway ($G_{i}$) stores $RID_{j}$ and $D_{j}$ for each robotic arm ($RM_{j}$). An attacker who has stolen the verifier table can impersonate the robotic arm ($RM_{j}$), as it obtains the secrets ($RID_{j},D_{j})$ of $RM_{j}$ and has the same ability as $RM_{j}$.

#### 2.2.3. Failure to Execute Correctly

In the scheme proposed by Kamil et al., the surgeon ($S_{k}$) cannot correctly compute $A_{6}=(R_{k}\|A_{5})⊕A_{3}$ in Step 1. Because $\left(R_{k}\|A_{5}\right)$ is longer than $A_{3}$, where $A_{3}=h(C,D_{i})$ and $A_{5}=h\left(R_{k},A_{3},TS_{1}\right)$, $S_{k}$ cannot directly execute an exclusive OR operation of $\left(R_{k}\|A_{5}\right)$ and $A_{3}$. Similar problems also occur in that $G_{i}$ cannot correctly compute $A_{8}=D_{j}⊕\left(R_{i}\;\left|\left|\;R_{k}^{*}\;\right|\right|TS_{2}\right)$ in Step 2, $RM_{j}$ cannot correctly compute $A_{11}=R_{i}^{*}⊕(R_{j}\;\|\;TS_{3})$ in Step 3, and $G_{i}$ cannot correctly compute $A_{13}=R_{i}\left|\left|R_{j}^{*}\right|\right|TS_{4}⊕R_{k}^{*}$ in Step 4.

## 3. Enhanced Authenticated Key Agreement Scheme for Tactile Internet Environment

In this section, we develop an enhanced AKA scheme based on the AKA scheme proposed by Kamil et al. for the Tactile Internet environment. In order to overcome the limitations of the AKA scheme proposed by Kamil et al., the enhanced scheme adopts a one-time key to protect communication messages such that an attacker who captures the robotic arm cannot derive valuable information from previous messages to perform impersonation attacks. To avoid stolen verifier attacks, $G_{i}$ does not directly store the secret key ($D_{j}$) of $RM_{j}$ in its database and protects $D_{j}$ with the secret key ($D_{i}$) of $G_{i}$. Even if the attacker steals the verification table, he/she still cannot obtain the secret key ($D_{j}$) of $RM_{j}$ to successfully impersonate $RM_{j}$.

A number of phases are involved in the enhanced scheme, including registration of gateways and robotic arms, registration of remote surgeons, login of remote surgeons, authentication and key agreement, updating of passwords, adding dynamic robotic arms, and revocation. Because the password updating phase, dynamic robotic arm addition phase, and revocation phase of the enhanced scheme are similar to the scheme proposed by Kamil et al., they are not discussed here. Below, we provide a detailed description of the gateway and robotic arm registration phase, the remote surgeon registration phase, the remote surgeon login phase, the authentication phase, and the key agreement phase. Figure 2 shows a flow chart of the enhanced scheme.

### 3.1. Registration Phase of Gateway and Robotic Arms

This phase provides the registration process for the gateway and robotic arms with the *TA*, as shown in Figure 3. The registration process is as follows.

Step 1: $\;TA\Rightarrow G_{i}:\;M_{1}=\left(RID_{i},D_{i},RID_{j},D_{j}\right)$.

The trust authority (TA) at first chooses a unique identity ($RID_{TA}$) and a one-way hash function operation ($h:{\left\{0,1\right\}}^{*}\to Z_{q}^{*}$). Next, the TA chooses $RID_{i}$ and $RID_{j}$ as the identities of the gateway ($G_{i}$) and the robotic arm ($RM_{j}$), respectively, picks a secret ($s\in Z_{q}^{*}$), and computes $D_{i}=h\left(s,RID_{TA},RID_{i}\right)$ and $D_{j}=h\left(s,RID_{TA},RID_{j}\right)$. Finally, the TA stores $\left(RID_{i},D_{i},RID_{j},D_{j}\right)$ and sends $M_{1}$ to $G_{i}$ through a secure channel.

Step 2: Gi⇒RMj: M2=(RIDj,Dj).

After the gateway ($G_{i}$) receives $M_{2}$, $G_{i}$ computes $CD_{j}=h(RID_{j}\;\|\;D_{i})\;⊕\;D_{j}$ and stores $\left(RID_{i},D_{i},RID_{j},CD_{j}\right)$. Finally, $G_{i}$ sends $M_{2}$ to $RM_{j}$.

### 3.2. User Registration Phase

In this phase, the remote surgeon ($S_{k}$) registers with the trusted authority (TA). Each surgeon ($S_{k}$) has a smart card with the information of the surgeon. The registration process of the remote surgeon is shown in Figure 4.

Step 1: $S_{k}\Rightarrow TA:\;M_{1}=\left(RID_{k},D_{k},HPW_{k}\right)$.

The remote surgeon ($S_{k}$) first picks his/her own identity ($RID_{k}$), password ($PW_{k}$), and a random number $B_{k}$ and computes $D_{k}=h\left(RID_{k},B_{k}\right)$ and $HPW_{k}=h\left(PW_{k},B_{k}\right)$. Finally, $S_{k}$ sends $M_{1}$ to the TA through a secure channel.

Step 2: $TA\Rightarrow S_{k}:M_{2}=\left(TID_{k},\alpha ,h\left(.\right)\right)$.

After receiving $M_{1}$, the TA first picks a random identity ($TID_{k}$) and computes $\alpha =h\left(TID_{k},D_{i}\right)⊕h\left(D_{k},HPW_{k}\right)$. Then, the TA stores (α,$\;TID_{k}$) in the memory of a mobile device and sends it to $S_{k}$ through a secure channel. Upon receiving the mobile device, $S_{k}$ computes $A_{1}=h\left(PW_{k},RID_{k}\right)⊕B_{k}$ and the verification message, $VM_{1}=h\left(B_{k},HPW_{k},D_{k}\right)$. Then, $S_{k}$ stores $A_{1}$, $VM_{1}$, $TID_{k}$, and α in the smart card.

### 3.3. Login, Authentication, and Session Key Agreement Phase

In order to perform remote operations in case of an emergency, the remote surgeon ($S_{k}$) needs to log in to a smart card and send a verification message to access the gateway ($G_{i}$). The gateway ($G_{i}$) sends a verification message to the robot after the remote surgeon has been identified. The robot passes the authentication message to the remote surgeon via the gateway. Finally, the gateway, remote coverage, and robotic arm establish a session key for the current login session. The authentication and key agreement of the proposed protocol is shown in Figure 5, and the details are summarized below.

Step 1: $S_{k}\to G_{i}:\;M_{1}=\left(TID_{k},A_{3},VM_{2},TS_{1}\right)$.

The remote surgeon ($S_{k}$) inputs his/her $RID_{k}$ and $PW_{k}$ into the mobile device; then, mobile device computes $B_{k}=A_{1}⊕h\left(RID_{k},PW_{k}\right)$ to obtain the random number ($B_{k}$) and computes $D_{k}=h\left(RID_{k},B_{k}\right)$, $HPW_{k}=h\left(PW_{k},B_{k}\right)$, and $VM_{1}^{*}=h\left(B_{k},HPW_{k},D_{k}\right)$ to verify $VM_{1}^{*}=?VM_{1}$. If successful, the mobile device picks the current timestamp ($TS_{1}$) and a random number ($R_{k}$) and computes $A_{2}=\alpha ⊕h\left(D_{k},HPW_{k}\right)$ and $A_{3}=h\left(A_{2},HPW_{k}\right)⊕R_{k}$ and verification the message, $VM_{2}=h\left(R_{k},A_{2},TS_{1}\right)$. Finally, $S_{k}$ sends $M_{1}$ to the gateway ($G_{i}$).

Step 2: $G_{i}\to RM_{j}:\;M_{2}=\left(TID_{k},A_{4},A_{5},VM_{3},TS_{2}\right)$.

When $G_{i}$ receives $M_{1}$, $G_{i}$ checks whether the timestamp ($TR_{1}-TS_{1}$) is less than ΔT. If successful, $G_{i}$ computes $A_{2}^{*}=h\left(TID_{k},D_{i}\right)$, $R_{k}^{*}=A_{3}⊕h\left(A_{2}^{*},TS_{1}\right)$, and $VM_{2}^{*}=h\left(R_{k}^{*},A_{2}^{*},TS_{1}\right)$ to verify $VM_{2}^{*}=?VM_{2}$. If successful, $G_{i}$ picks a random number ($R_{i}$) and the current timestamp ($TS_{2}$) and computes $D_{j}=h(RID_{j}\;\|\;D_{i})\;⊕\;CD_{j}$ to obtain the $D_{j}$ of $RM_{j}$, then computes $A_{4}=h\left(D_{j},TS_{2},0\right)⊕R_{i}$, $A_{5}=h\left(D_{j},TS_{2},1\right)⊕R_{k}^{*}$, and a verification message, $VM_{3}=h\left(RID_{j},D_{j},TID_{k},R_{i},TS_{2}\right)$, where $D_{j}$ is the secret of the robotic arm, and $TS_{2}$ ensures the freshness of messages.

Step 3: $RM_{j}\to G_{i}:\;M_{3}=\left(A_{6},VM_{4},TS_{3}\right)$.

After receiving $M_{2}$ from $G_{i}$, $RM_{j}$ checks whether the timestamp ($TR_{2}-TS_{2}$) is less than ΔT. If successful, $RM_{j}$ computes $R_{i}^{*}=A_{4}⊕h\left(D_{j},TS_{2},0\right)$, $R_{k}^{**}=A_{5}⊕h\left(D_{j},TS_{2},1\right)$, and $VM_{3}^{*}=h\left(RID_{j},D_{j},TID_{k},R_{i}^{*}TS_{2}\right)$ to verify $VM_{3}^{*}=?VM_{3}$. If successful, $RM_{j}$ picks a random number ($R_{j}$) and the current timestamp ($TS_{3}$) and computes the session key ($K_{1}=h\left(R_{i}^{*},R_{k}^{**},R_{j}\right)$), $A_{6}=h\left(R_{i}^{*},TS_{3}\right)⊕R_{j}$, and the verification message ($VM_{4}=h\left(R_{i}^{*},R_{j},K_{1},RID_{j},D_{j},TS_{3}\right)$). Then, $RM_{j}$ sends $M_{3}$ to $G_{i}$.

Step 4: $G_{i}\to S_{k}:\;M_{4}=\left(A_{7},A_{8},A_{9},VM_{5},TS_{4}\right)$.

When $G_{i}$ receives $M_{1}$, $G_{i}$ checks whether the timestamp ($TR_{3}-TS_{3}$) is less than ΔT. If successful, $G_{i}$ computes $R_{j}^{*}=A_{6}⊕h\left(R_{i},TS_{3}\right)$, $K_{2}=h\left(R_{i},R_{k}^{*},R_{j}^{*}\right)$, and the verification message ($VM_{4}^{*}=h\left(R_{i},R_{j}^{*},K_{2},RID_{j},D_{j},TS_{3}\right)$) to verify $VM_{4}^{*}=?VM_{4}$. If successful, $G_{i}$ picks the current timestamp ($TS_{4}$) and computes $A_{7}=h\left(A_{2}^{*},TS_{4},0\right)⊕R_{i}$, $A_{8}=h\left(A_{2}^{*},TS_{4},1\right)⊕R_{j}^{*}$, $TID_{k}^{new}=h\left(A_{2}^{*},K_{2}\right)$, $A_{2}^{new}=h\left(TID_{k}^{new},D_{i}\right)$, $A_{9}=h\left(A_{2}^{*},TS_{4},2\right)⊕A_{2}^{new}$, and $VM_{5}=h\left(K_{2},A_{2}^{new},TS_{4}\right)$. Finally, $G_{i}$ sends $M_{4}$ to $S_{k}$.

Step 5: Update ${\mathrm{TID}}_{k}$ and α in $S_{k}$.

After $S_{k}$ receives $M_{4}$, $S_{k}$ checks whether the timestamp ($TR_{4}-TS_{4}$) is less than ΔT. If successful, $S_{k}$ computes $R_{i}^{*}=h\left(A_{2}^{*},TS_{4},0\right)⊕A_{7}$ and $R_{j}^{**}=h\left(A_{2}^{*},TS_{4},1\right)⊕A_{8}$ to obtain the random number ($R_{i}^{*}$) of $G_{i}$ and the random number ($R_{j}^{**}$) of $RM_{j}$. Next, $S_{k}$ computes the session key ($K_{3}=h\left(\;R_{i}^{*},\;R_{j}^{**},R_{k}\right)$), $A_{2}^{new}=A_{9}⊕h\left(A_{2},TS_{4},2\right)$, and $TID_{k}^{new}=h\left(A_{2},K_{3}\right)$. Then, $S_{k}$ computes $VM_{5}^{*}=h\left(K_{3},A_{2}^{new},TS_{4}\right)$ to verify $VM_{5}^{*}=?VM_{5}$. If successful, $S_{k}$ computes $\alpha^{new}=A_{2}^{new}⊕h\left(D_{k},HPW_{k}\right)$ and updates α and $TID_{k}$ via $\alpha^{new}$ and $TID_{k}^{new}$ in the smart card.

## 4. Security and Performance Analysis

An analysis and comparison of the performance and security of the enhanced scheme are provided in this section.

### 4.1. Authentication Proof of the Proposed Scheme Using BAN Logic

BAN logic [13] is used in this subsection to verify that the proposed scheme satisfies the session key security and mutual authentication requirements. Table 2 lists the notations of BAN logic.

#### 4.1.1. Inference Rules of BAN Logic

Below, we present a list of the rules and logical postulates of BAN logic [13].

**Rule 1.** P|≡P ↔K Q, P ◁ 〈X〉K P|≡Q|~X: If entity P believes that secret K is shared with Q and sees message X is encrypted using K, then P believes that Q once said X.

**Rule 2.** $\frac{P\left|\equiv \#\left(X\right),\;P\right|\equiv Q|\sim X\;}{P\left|\equiv Q\right|\equiv X}$: If entity P believes that X is fresh and entity Q once said X, then P believes that Q believes X.

**Rule 3.** $\frac{P\left|\equiv Q⟹X,\;P\right|\equiv Q|\equiv X\;}{P|\equiv X}$: If entity P believes that Q has jurisdiction over X and Q believes X, then P believes that X is true.

**Rule 4.** P|≡#(X), P|≡Q|≡X P|≡P ↔K Q: If entity P believes that X is fresh and Q believes X, then P believes secret K that is shared between entities P and Q.

**Rule 5.** $\frac{P|\equiv \#\left(X\right)}{P|\equiv \#\left(X,\;Y\right)}$: If entity P believes that X is fresh, then P believes in the freshness of (X, Y).

#### 4.1.2. Goals of Authentication and Key Agreement

In this subsection, we demonstrate that the proposed scheme satisfies the following goals to ensure its security according to the above assumptions and postulates.

**Goal 1**: Gi |≡Sk |≡Gi ↔K Sk.

**Goal 2**: Gi |≡RMj|≡Gi ↔K RMj.

**Goal 3:** RMj |≡Gi |≡ RMj  ↔K Gi.

**Goal 4**: Sk |≡Gi |≡ Sk  ↔K Gi.

**Goal 5**: Sk |≡ RMj|≡Sk ↔K RMj.

**Goal 6**: RMj|≡Sk |≡ RMj ↔K Sk.

#### 4.1.3. Idealized Form

The proposed scheme is transformed into an idealized form in the following manner.

**M_1_**. $\left(S_{k}\to \;G_{i}\right):\;TID_{k},A_{3}:{\langle R_{k}\rangle }_{h(A_{2},TS_{1})},VM_{2}:h(R_{k},A_{2},TS_{1}),TS_{1}$.

**M_2_.** $\left(G_{i}\to \;RM_{j}\right):\;TID_{k},A_{4}:{\langle R_{i}\rangle }_{h(D_{j},TS_{2},0)},A_{5}:{\langle R_{k}^{*}\rangle }_{h(D_{j},TS_{2},1)},$

$VM_{3}:h(RID_{j},D_{j},TID_{k},R_{i},TS_{2},R_{k}^{*}),TS_{2}$.

**M_3_.** ($RM_{j}\to \;G_{i}):\;A_{6}:{\langle R_{j}\rangle }_{h\left(R_{i}^{*},TS_{3}\right)},VM_{4}:h(K_{1},RID_{j},D_{j},TS_{3}),TS_{3}$.

**M_4_.** ($G_{i}\to \;S_{k}):A_{7}:{\langle R_{i}\rangle }_{h\left(A_{2}^{*},TS_{4},0\right)},A_{8}:{\langle R_{j}^{*}\rangle }_{h\left(A_{2}^{*},TS_{4},1\right)},A_{9}:{\langle A_{2}^{new}\rangle }_{h\left(A_{2}^{*},TS_{4},2\right)},$

$VM_{5}:h\left(K_{2},A_{2}^{new},TS_{4}\right),TS_{4}$.

#### 4.1.4. Assumptions

According to the following assumptions, in this subsection, we prove that the proposed scheme satisfies the security properties.

$AS_{1}:\;G_{i}$*|*≡*#*$\;h(R_{k},A_{2},TS_{1})$.

$AS_{2}:\;G_{i}$*|*≡*#*$\;h(K_{1},RID_{j},D_{j},TS_{3})$.

$AS_{3}:\;G_{i}$*|*$\equiv \;G_{i}\;\overset{\;A_{2}:\;h(TID_{k},D_{i})\;}{↔}\;S_{k}$.

$AS_{4}:\;S_{k}$*|*$\equiv \;S_{k}\;\overset{\;A_{2}:\;h(TID_{k},D_{i})\;}{↔}\;G_{i}$.

$AS_{5}:\;G_{i}$*|*$\equiv \;G_{i}\;\overset{\;D_{j}\;}{↔}\;RM_{j}$.

$AS_{6}:\;RM_{j}$*|*$\equiv \;RM_{j}\;\overset{\;D_{j}\;}{↔}\;G_{i}$.

$AS_{7}:\;RM_{j}$*|*≡*#*$\;h(RID_{j},D_{j},TID_{k},R_{i},TS_{2},R_{k}^{*})$.

$AS_{8}:\;S_{k}$*|*≡*#*$\;h\left(K_{2},A_{2}^{new},TS_{4}\right)$.

$AS_{9}:\;S_{k}$*|*$\equiv \;G_{i}\;⟹\;R_{i}$.

$AS_{10}:\;S_{k}$*|*$\equiv \;RM_{j}\;⟹\;R_{j}$.

$AS_{11}:\;G_{i}$*|*$\equiv \;S_{k}\;⟹R_{k}$.

$AS_{12}:\;G_{i}$*|*$\equiv \;RM_{j}\;⟹R_{j}$.

$AS_{13}:\;RM_{j}$*|*$\equiv \;G_{i}\;⟹\;R_{i}$.

$AS_{14}:\;RM_{j}$*|*$\equiv \;S_{k}\;⟹\;R_{k}$.

#### 4.1.5. Verification

Based on the above assumptions and the logic of BAN, the following confirms the correctness of the proposed scheme. By using Message **M_1_**,

$G_{i}\;◁$ {$TID_{k},A_{3}:{\langle R_{k}\rangle }_{h(A_{2},TS_{1})},VM_{2}:h(R_{k},A_{2},TS_{1}),TS_{1}$}.

From Rule 1 and $AS_{3}$,

$V_{1}$: $G_{i}$ |$\equiv \;S_{k}$ |$\sim R_{k}$.

From Rule 2 and $AS_{1}$,

$V_{2}$: $G_{i}$ |$\equiv \;S_{k}$ |$\equiv \;R_{k}$.

Then, from Rule 3 and $AS_{11}$,

$V_{3}$: $G_{i}$ |$\equiv \;R_{k}$.

According to Rule 4, $AS_{1}$ and $V_{2}$,

$V_{4}$: $G_{i}$ |≡ Gi  ↔K Sk.

Further, using Rule 2, $AS_{1}$ and $V_{1}$,

$V_{5}$: $G_{i}$ |$\equiv \;S_{k}$ |≡ Gi  ↔K Sk.           **Goal 1**

Similarly, by using Message **M_3_**,

$G_{i}\;◁$ {$A_{6}:{\langle R_{j}\rangle }_{h\left(R_{i}^{*},TS_{3}\right)},VM_{4}:h(K_{1},RID_{j},D_{j},TS_{3}),TS_{3}$}.

From Rule 1 and $AS_{5}$,

$V_{6}$: $G_{i}$ |$\equiv \;RM_{j}$ |$\sim R_{j}$.

From Rule 2 and $AS_{2}$ and $V_{6}$,

$V_{7}$: $G_{i}$ |$\equiv \;RM_{j}$ |$\equiv \;R_{j}$.

From Rule 3 and $AS_{12}$,

$V_{8}$: $G_{i}$ |$\equiv \;R_{j}$.

According to Rule 4, $AS_{2}$ and $V_{7}$,

$V_{9}$: $G_{i}$ |≡ Gi  ↔K RMj.

Using Rule 2, $AS_{2}$ and $V_{6}$, we have

$V_{10}$: $G_{i}$ |$\equiv \;RM_{j}$ |≡ Gi  ↔K RMj.        **Goal 2**

By using Message **M_2_**,

$RM_{j}$ ◁ {$TID_{k},A_{4}:\langle R_{i}\rangle {}_{h(D_{j},TS_{2},0)},A_{5}:\langle R_{k}^{*}\rangle {}_{h(D_{j},TS_{2},1)},$

$VM_{3}:h(RID_{j},D_{j},TID_{k},R_{i},TS_{2},R_{k}^{*}),TS_{2}$}.

From Rule 1 and AS6,

$V_{11}$: $RM_{j}$ |$\equiv G_{i}\;$|$\sim R_{i}$.

From Rule 2 and AS7,

$V_{12}$: $RM_{j}$ |$\equiv G_{i}$ |$\equiv \;R_{i}$.

Then, from Rule 3 and $AS_{13}$,

$V_{13}$: $RM_{j}$ |$\equiv \;R_{i}$.

According to Rule 4, $AS_{7}$ and $V_{12}$,

$V_{14}$: $RM_{j}$ |≡ RMj  ↔K Gi.

Further, using Rule 2, $AS_{7}$ and $V_{11}$,

$V_{15}$: $RM_{j}$ |$\equiv \;G_{i}$ |≡ RMj  ↔K Gi.        **Goal 3**

Similarly, by using Message **M_4_,**

$S_{k}$ ◁ {$A_{7}:\langle R_{i}\rangle {}_{h\left(A_{2}^{*},TS_{4},0\right)},A_{8}:\langle R_{j}^{*}\rangle {}_{h\left(A_{2}^{*},TS_{4},1\right)},A_{9}:\langle A_{2}^{new}\rangle {}_{h\left(A_{2}^{*},TS_{4},2\right)},$

$VM_{5}:h\left(K_{2},A_{2}^{new},TS_{4}\right),TS_{4}$}.

From Rule 1 and AS6,

$V_{16}$: $S_{k}$ |$\equiv G_{i}\;$|$\sim R_{i}$.

From Rule 2 and $AS_{8}$,

$V_{17}$: $S_{k}$ |$\equiv G_{i}$ |$\equiv \;R_{i}$.

Then, from Rule 3 and $AS_{9}$,

$V_{18}$: $S_{k}$ |$\equiv \;R_{i}$.

According to Rule 4, $AS_{8}$ and $V_{17}$,

$V_{19}$: $S_{k}$ |≡ Sk  ↔K Gi.

Further, using Rule 2, $AS_{8}$ and $V_{16}$,

$V_{20}$: $S_{k}$ |$\equiv \;G_{i}$ |≡ Sk  ↔K Gi.           **Goal 4**

By using Message **M_4_,**

$V_{21}$: $S_{k}$ |$\equiv RM_{j}$ |$\sim \;R_{j}$.

From Rule 2 and $AS_{2}$,

$V_{22}$: $S_{k}$ |$\equiv RM_{j}$ |$\equiv \;R_{j}$.

Then, from Rule 3 and $AS_{10}$,

$V_{23}$: $S_{k}$ |$\equiv \;R_{j}$.

According to Rule 4, $AS_{2}$ and $V_{22}$,

$V_{24}$: Sk |≡ Sk ↔KRMj.

Further, using Rule 2, $AS_{2}$ and $V_{21}$,

$V_{25}$: $S_{k}$ |≡RMj |≡ Sk ↔KRMj.         **Goal 5**

By using Message **M_2_**,

$V_{26}$: $RM_{j}$ |$\equiv S_{k}\;$|$\sim R_{k}$.

From Rule 2 and AS7,

$V_{27}$: $RM_{j}$ |$\equiv S_{k}$ |$\equiv \;R_{k}$.

Then, from Rule 3 and $AS_{14}$,

$V_{28}$: $RM_{j}$ |$\equiv \;R_{k}$.

According to Rule 4, $AS_{7}$ and $V_{27}$,

$V_{29}$: $RM_{j}$ |≡ RMj  ↔K Sk.

Further, using Rule 2, $AS_{7}$ and $V_{26}$,

$V_{30}$: $RM_{j}$ |$\equiv \;S_{k}$ |≡ RMj  ↔K Sk.        **Goal 6**

The proof is concluded.

### 4.2. Security Analysis

The security requirements of the enhanced scheme are discussed in this subsection. The enhances scheme uses the properties of the scheme proposed by Kamil et al. [9]. The arguments of some security requirements, including provision of strong anonymity; session key establishment; perfect forward secrecy; and resistance to replay attacks, impersonation attacks, offline user login credentials guessing attacks, insider attacks, mobile device loss attacks, and denial of service attacks, are similar to those in the scheme proposed by Kamil et al. and are therefore not discussed here. These security requirements include resistance to robotic arm compromise attacks and resistance to stolen verifier table attacks, as described below.

#### 4.2.1. Resistance to Robotic Arm Compromise Attacks

In the enhanced scheme, even if the attacker compromises the robotic arm ($RM_{j}$) and obtains ($RID_{j}$,$\;D_{j}$) from $RM_{j}$, the attacker cannot indirectly obtain information about remote surgeons and the gateway ($G_{i}$). Additionally, because the ($RID_{j}$,$\;D_{j}$) of each robotic arm is independent, as destroying a robotic arm, the attacker can communicate with $S_{k}$, but it does not affect the security of $S_{k}$’s communication with other robotic arms. The same is true for the gateway. Therefore, the proposed scheme is resilient against robot compromise attack.

#### 4.2.2. Resistance to Stolen Verifier Attacks

In the enhanced scheme, the gateway ($G_{i}$) stores ($RID_{j}$,$\;CD_{j}$) instead of ($RID_{j}$,$\;D_{j}$), where $CD_{j}=D_{j}⊕\;h(RID_{j}\;\|\;D_{i})$, $D_{j}$ is the secret key of $RM_{j}$, and $D_{i}$ is the secret key of $G_{i}$. The verifier table does not contain $G_{i}$’s secret key ($D_{i}$). Then, an attacker who has stolen the verifier table cannot derive $D_{j}$ from ($RID_{j}$,$\;CD_{j}$) without $D_{i}$, and it is difficult to impersonate $RM_{j}$. Therefore, the enhanced scheme is resilient against stolen verifier table attacks.

### 4.3. Functionality Comparison

Table 3 compares the enhanced AKA scheme with related AKA schemes in term of security functionality. The enhanced AKA scheme provides more security requirements than related AKA schemes and is secure against potential attacks. Furthermore, it can resist robotic arm compromise attacks and stolen verifier table attacks.

### 4.4. Performance Comparisons

Table 4 shows comparisons between the enhanced AKA scheme and related AKA schemes in terms of computational cost, where $T_{h}$ denotes the execution time of a one-way hash function, $T_{e}$ denotes the execution time of a point multiplication based on ECC, and $T_{f}$ denotes the execution time of a fuzzy extractor. The experiment is run on an Intel CPU i3-3220 3.3 Ghz, RAM 4096 MB, Windows 7 Professional 64-bit, Eclipse Java Mars and Java SE 1.8. The hash function uses SHA-1, the point multiplication is based on ECC with a 16-bit key, and the fuzzy extractor refers to [11,17].

The scheme proposed by Kamil et al. [11] requires 20 hash operations, the scheme proposed by Amin et al. [5] requires 37 hash operations, the scheme proposed by Wu et al. [6] requires 34 hash operations, the scheme proposed by Chandrakar [7] requires 29 hash operations, the scheme proposed by Guo et al. [14] requires 36 hash operations, and our enhanced scheme requires 35 hash operations. The scheme proposed by Soni et al. [15] requires 31 hash operations, 6 point multiplications based on ECC, and 11 fuzzy extractor operations. The scheme proposed by Li et al. [16] requires 20 hash operations and 8 point multiplications based on ECC. Both these schemes ([15,16]) require time-consuming point multiplications based on ECC. The enhanced AKA scheme adopts a one-time key to protect communication messages and protects the verifier table with the *G_i_*’s secret key, so it requires more computations and response time than the AKA protocol proposed by Kamil et al. However, the enhanced AKA scheme addresses the limitations of the scheme proposed by Kamil et al., providing improved functionality while retaining a low computational cost.

## 5. Conclusions

In this paper, we addressed the limitations of the AKA scheme proposed by Kamil et al. for a Tactile Internet environment, including failure to resist robotic arm compromise attacks, failure to resist stolen verifier attacks, and failure to execute correctly. In order to address these limitations, an enhanced AKA scheme based the scheme proposed by Kamil et al. was developed by adopting a one-time key to protect communication messages and protecting the verifier table with a gateway secret key. Although the enhanced scheme requires more computations than the AKA protocol proposed by Kamil et al. it retains a low computational cost and provides more security features. Therefore, the enhanced AKA scheme is suitable for the Tactile Internet environment.

## Figures and Tables

**Figure 1 sensors-22-07941-f001:**
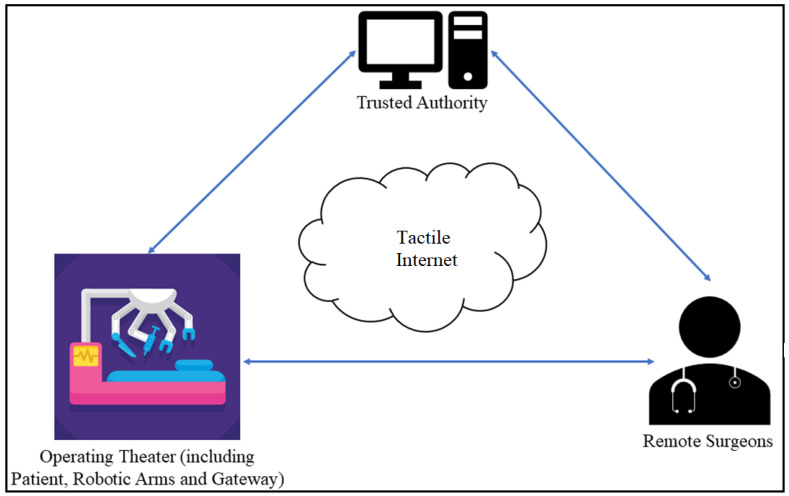
A simple model of a Tactile Internet remote surgery application.

**Figure 2 sensors-22-07941-f002:**
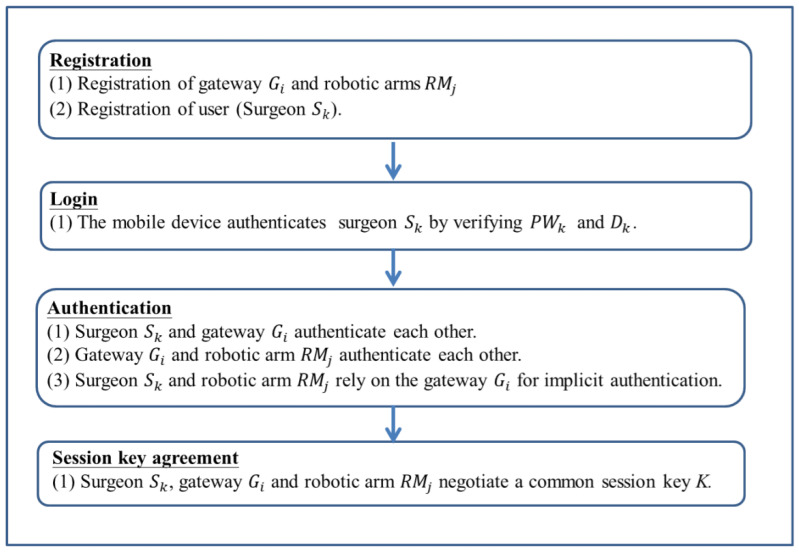
Flow chart of the enhanced scheme.

**Figure 3 sensors-22-07941-f003:**
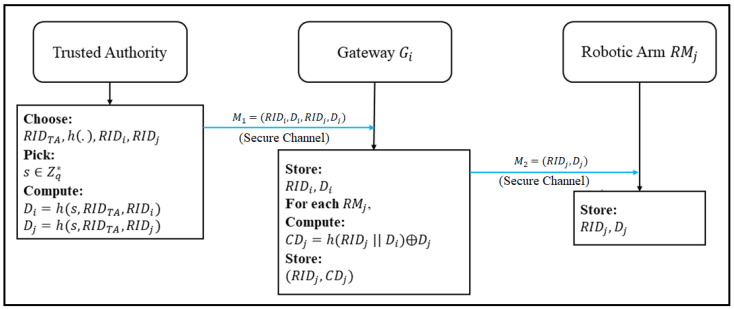
Registration process of gateway and robotic arms of the enhanced scheme.

**Figure 4 sensors-22-07941-f004:**
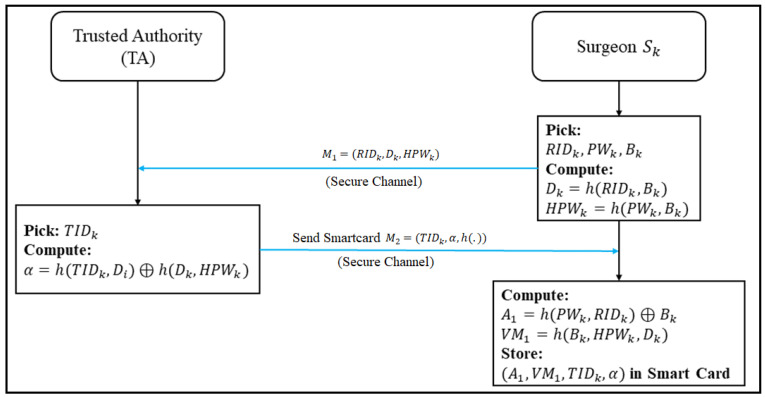
Registration phase of the remote surgeon of the proposed scheme.

**Figure 5 sensors-22-07941-f005:**
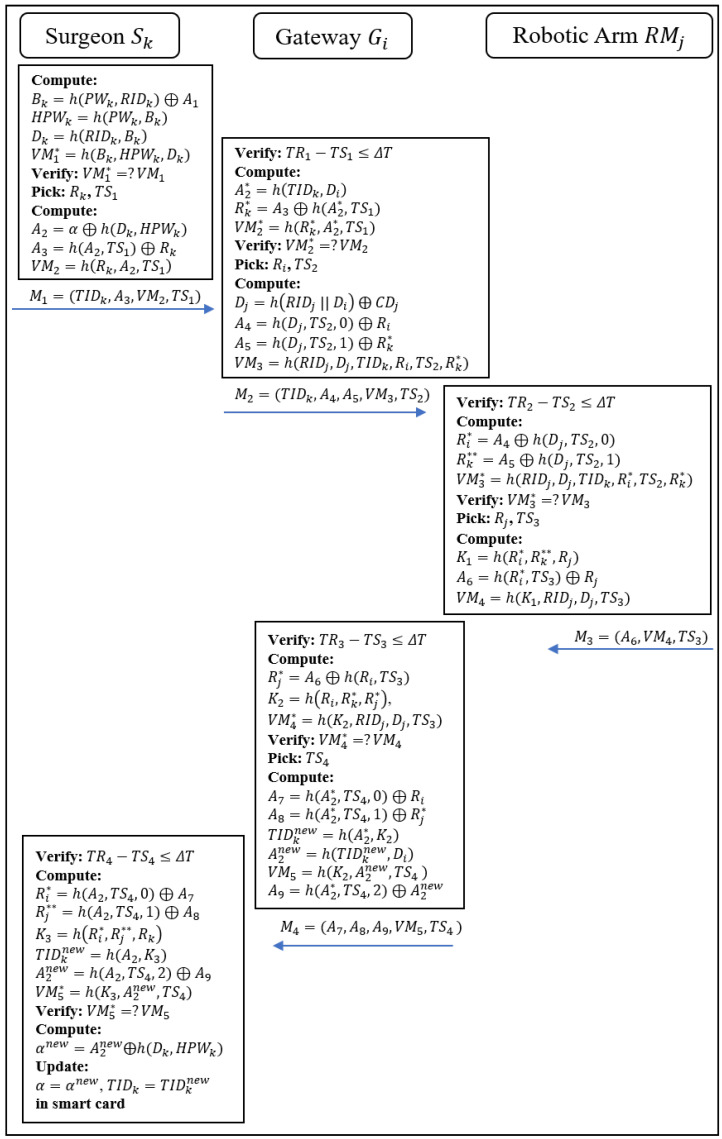
Login, authentication, and session key agreement phase of the enhanced scheme.

**Table 2 sensors-22-07941-t002:** BAN logic notations and respective abbreviations [13].

Notation	Abbreviation
P |≡ X	Entity P believes statement X
P ⟹X	P has jurisdiction over statement X
P |~ X	P once said X
P ◁ X	P sees X
${\langle X\rangle }_{K}$	Formula X is encrypted by key K
$P\overset{\;K\;}{↔}Q$	P and Q communicate via shared key K
P→ Q:m	P sends the message (m), and Q receives it
#X	Message #X is freshly generated

**Table 3 sensors-22-07941-t003:** Functionality comparisons.

Security Attribute	[11]	[5]	[6]	[7]	[14]	[15]	[16]	Our AKA
Provision of strong anonymity	O	O	X	O	X	O	O	O
Provision of session key establishment	O	-	O	-	O	O	O	O
Provision of perfect forward secrecy	O	O	O	O	O	O	O	O
Resistance to replay attacks	O	X	X	O	O	O	X	O
Resistance to impersonation attacks	O	X	O	O	O	O	O	O
Resistance to offline user login credentials guessing attack	O	X	O	O	O	O	O	O
Resistance to insider attacks	O	-	O	O	O	O	O	O
Resistance to mobile device loss attacks	O	X	O	O	O	O	O	O
Resistance to denial of service attacks	O	O	O	O	O	O	O	O
Resistance to robotic arm compromise attacks	X	X	O	O	O	O	O	O
Resistance to stolen verifier attacks	X	O	O	X	O	X	X	O

O: the property is satisfied, X: the property is not satisfied; -: the property is not considered.

**Table 4 sensors-22-07941-t004:** Computation cost comparison.

Scheme	Mobile Device/User	Gateway	Sensor Node/Robotic Arm	Total/Response Time
[11]	$8T_{h}$	$8T_{h}$	$4T_{h}$	$20T_{h}$/240 ms.
[5]	$12T_{h}$	$19T_{h}$	$6T_{h}$	$37T_{h}$/444 ms.
[6]	$11T_{h}$	$17T_{h}$	$6T_{h}$	$34T_{h}$/408 ms.
[7]	$11T_{h}$	$13T_{h}$	$5T_{h}$	$29T_{h}$/348 ms.
[14]	$13T_{h}$	$17T_{h}$	$6T_{h}$	$36T_{h}$/432 ms.
[15]	$13T_{h}+3T_{e}+13T_{f}$	$11T_{h}+3T_{e}$	$7T_{h}$	$31T_{h}+6T_{e}+13T_{f}$/1645 ms.
[16]	$8T_{h}+3T_{e}$	$8T_{h}+3T_{e}$	$4T_{h}+2T_{e}$	$20T_{h}+8T_{e}$/776 ms.
Our AKA	$13T_{h}$	$16T_{h}$	$6T_{h}$	$35T_{h}$/420 ms.

## Data Availability

Not applicable.

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
