# Peer review of "Enhanced Authenticated Key Agreement for Surgical Applications in a Tactile Internet Environment"

_sensors, 2022, doi:10.3390/s22207941_

Round 1

Reviewer 1 Report

1. Since the article is only about authentication in a surgical application of Tactile internet, I would suggest adding the word "medical" or "surgical" in the title, or explaining in the introduction how the proposed scheme can be applied to other Tactile internet applications.

2. Keywords could be better worded.

3. I would sugggect renaming "Review of literatures" to "Literature rewiev" or "Related works"

4. In lines 73 и 75 authors apparently meant "ms" instead of "m/s"

5. The literature review includes articles that describe security issues in medical sensor systems, and not in medical Tactile internet systems, which may not be entirely relevant to the topic of the article. 

6. In lines 185 and 186, instead of links to sources, an error is printed.

7. The font in which the links are written may be too large.

Author Response

Reply to the Reviewer’s Comments

Title: "Enhanced Authenticated Key Agreement for Tactile Internet Environment"

Authors: Tian-Fu Lee, Xiucai Ye, Wei-Yu Chen and Chi-Chang Chang

Reference No.: sensors-1950808

Summary of the changes in response to the review of sensors-1950808

We would like to thank the editor and the anonymous reviewers for your very helpful and valuable comments on our manuscript entitled "Enhanced Authenticated Key Agreement for Tactile Internet Environment". We have revised the manuscript according to the reviewers’ comments. The main changes in response to the reviewing results are listed below.

  1. We have revised the title, Keywords and a subtitle according to the reviewers’ comments.
  2. We have revised the descriptions of related works according to the reviewers’ comments.
  3. The typo errors and mistakes have been corrected and revised in this version according to the reviewers’ comments.

Reviewer 1 Comments

1.Since the article is only about authentication in a surgical application of Tactile internet, I would suggest adding the word "medical" or "surgical" in the title, or explaining in the introduction how the proposed scheme can be applied to other Tactile internet applications.

Ans.: We thank the reviewer for this constructive suggestion. We have revised the title as “Enhanced Authenticated Key Agreement for Surgical Applications in Tactile Internet Environment“ according to the reviewers’ comments.

2.Keywords could be better worded.

Ans.: We thank the reviewer for this constructive suggestion. We have revised the Keywords according to the reviewers’ comments.

3.I would suggest renaming "Review of literatures" to "Literature review" or "Related works"

Ans.: We thank the reviewer for this constructive suggestion. We have revised the subtitle as "Related works" according to the reviewers’ comments. The descriptions are shown on Page 2, Sec. 1.2, Line 71.

4.In lines 73 и 75 authors apparently meant "ms" instead of "m/s"

Ans.: We thank the reviewer for this constructive suggestion. We have corrected this typo error according to the reviewers’ comments. The descriptions are shown on Page 2, Sec. 1.2, Lines 74 and 76.

5.The literature review includes articles that describe security issues in medical sensor systems, and not in medical Tactile internet systems, which may not be entirely relevant to the topic of the article. 

Ans.: We thank the reviewer for this constructive suggestion. Indeed, the related works described in this paper include authenticated key agreement approaches for remote medical sensor systems. However, these authentication schemes and authentication schemes for tactile internet in remote surgery are close in structure. Additionally, recent references to authentication schemes for tactile internet in remote surgery are limited. Thus, recent authentication approaches for remote medical sensor systems are included. The descriptions are shown on Page 2, Sec. 1.2.

6.In lines 185 and 186, instead of links to sources, an error is printed.

Ans.: We thank the reviewer for this constructive suggestion. We have revised these mistakes according to the reviewers’ comments. The descriptions are shown on Page 5, Lines 190-191.

7.The font in which the links are written may be too large.

Ans.: We thank the reviewer for this constructive suggestion. We have revised the font according to the reviewers’ comments. The descriptions are shown on Figures 1-4 and 6-8.

Reviewer 2 Report

The reviewer comments for authors:

1.    The paper is well-organized and well-written, and ideas are coherent and arranged logically.

2.    The methodology, result and discussion sections are described clearly.

3.    The conclusions are supported by the obtained result.

4.    Few formatting mistakes are listed below:

·      Lines 185 and 186 have errors in referencing figures 2 and 3

·      In line 205, the authors referenced figure 1 by mistake, they need to reference figure 4.

·      In line 390, figure 5 is referenced in the text after its appearance.

·      In line 406, figure 6 is numbered as figure 5 by mistake.

·      Line 410 has error in referencing figure 7.

·      In line 411, write M2 in the figure instead of M4 to be the same symbol used in the text explanation.

·      In line 412, figure 7 is numbered as figure 6 by mistake.

·      In line 454, the figure should be numbered as figure 8.

·      Line 475 has error in referencing table 2.

Author Response

Reply to the Reviewer’s Comments

Title: "Enhanced Authenticated Key Agreement for Tactile Internet Environment"

Authors: Tian-Fu Lee, Xiucai Ye, Wei-Yu Chen and Chi-Chang Chang

Reference No.: sensors-1950808

Summary of the changes in response to the review of sensors-1950808

We would like to thank the editor and the anonymous reviewers for your very helpful and valuable comments on our manuscript entitled "Enhanced Authenticated Key Agreement for Tactile Internet Environment". We have revised the manuscript according to the reviewers’ comments. The main changes in response to the reviewing results are listed below.

  1. We have revised the title, Keywords and a subtitle according to the reviewers’ comments.
  2. We have revised the descriptions of related works according to the reviewers’ comments.
  3. The typo errors and mistakes have been corrected and revised in this version according to the reviewers’ comments.

Reviewer 2 Comments

  1. The paper is well-organized and well-written, and ideas are coherent and arranged logically.
  2. The methodology, result and discussion sections are described clearly.
  3. The conclusions are supported by the obtained result.
  4. Few formatting mistakes are listed below:

Ans.: We thank the reviewer for this constructive suggestion. The revisions and responses to comments are described below.

(1) Lines 185 and 186 have errors in referencing figures 2 and 3 Pages 5-6.

(2) In line 205, the authors referenced figure 1 by mistake, they need to reference figure 4.

Ans.: We thank the reviewer for this constructive suggestion. We have revised these mistakes according to the reviewers’ comments. The descriptions are shown on Pages 5-6, Page 5, Lines 190-191 and 210.

(3) In line 390, figure 5 is referenced in the text after its appearance.

 In line 406, figure 6 is numbered as figure 5 by mistake.

Line 410 has error in referencing figure 7.

 In line 412, figure 7 is numbered as figure 6 by mistake.

 In line 454, the figure should be numbered as figure 8.

Ans.: We thank the reviewer for this constructive suggestion. We have revised these mistakes according to the reviewers’ comments. The descriptions are shown on Pages 12-14, Figures 6-8.

(4)  In line 411, write M2 in the figure instead of M4 to be the same symbol used in the text explanation.

Ans.: We thank the reviewer for this constructive suggestion. We have revised these mistakes according to the reviewers’ comments. The descriptions are shown on Page 12, Figure 7.

(5) Line 475 has error in referencing table 2.

Ans.: We thank the reviewer for this constructive suggestion. We have revised this error according to the reviewers’ comments. The descriptions are shown on Page 15, Sec.4.1, Line 485.
